# Non-Invasive and Confirmatory Differentiation of Hermaphrodite from Both Male and Female Cannabis Plants Using a Hand-Held Raman Spectrometer

**DOI:** 10.3390/molecules27154978

**Published:** 2022-08-05

**Authors:** Nicolas K. Goff, James F. Guenther, John K. Roberts, Mickal Adler, Michael Dalle Molle, Greg Mathews, Dmitry Kurouski

**Affiliations:** 1Department of Biochemistry and Biophysics, Texas A&M University, College Station, TX 77843, USA; 2Mariposa Technology, Metairie, LA 70002, USA; 3Element 6 Dynamics, Longmont, CO 80504, USA

**Keywords:** cannabis, sex determination, Raman spectroscopy, chemometrics

## Abstract

Cannabis (*Cannabis sativa* L.) is a dioecious plant that produces both male and female inflorescences. In nature, male and female plants can be found with nearly equal frequency, which determines species out-crossing. In cannabis farming, only female plants are preferred due to their high yield of cannabinoids. In addition to unfavorable male plants, commercial production of cannabis faces the appearance of hermaphroditic inflorescences, species displaying both pistillate flowers and anthers. Such plants can out-cross female plants, simultaneously producing undesired seeds. The problem of hermaphroditic cannabis triggered a search for analytical tools that can be used for their rapid detection and identification. In this study, we investigate the potential of Raman spectroscopy (RS), an emerging sensing technique that can be used to probe plant biochemistry. Our results show that the biochemistry of male, female and hermaphroditic cannabis plants is drastically different which allows for their confirmatory identification using a hand-held Raman spectrometer. Furthermore, the coupling of machine learning approaches enables the identification of hermaphrodites with 98.7% accuracy, whereas both male and female plants can be identified with 100% accuracy. Considering the label-free, non-invasive and non-destructive nature of RS, the developed optical sensing approach can transform cannabis farming in the U.S. and overseas.

## 1. Introduction

*Cannabis sativa* L., which is also known as hemp and cannabis, is a dioecious diploid (2 n = 20) of *Cannabaceae* family [1,2]. Dioecy is a unique evolutionary phenomenon observed in only 6% of all angiosperm plant species [3]. Dioecious species possess only male or female inflorescences on different plants [4]. The appearance of both male and female plants is expressed at very early stages. This sexual dimorphism is controlled by microRNA, activation of sex-determining genes, and sex chromosomes, as well as DNA methylation [5,6]. The required cross-fertilization of such dioecious plants is an evolutionary mechanism that facilitates their genetic diversity and heterozygosity [7,8].

Female plants develop trichomes on flower bracts. These appendages contain cannabinoids, psychologically and physiologically active molecules that include delta-9-tetrahydrocannabinol (Δ-9-THC), cannabidiol (CBD), and cannabigerol (CBG) [9,10,11]. Trichomes are not formed on male plants. Therefore, male plants are eliminated in the early stages of their vegetation. In addition to male and female plants, hermaphroditic inflorescences can develop spontaneously in *Cannabis sativa* L. Such plants predominantly possess female inflorescences, but anthers (ranging from a few to many) can be also observed within the leaf axils or in pistillate flower buds [2,12]. The hermaphrodite plants are monoecious, allowing the production of both pollen and seeds [13]. Therefore, the presence of such plants on the farm can drastically alter the cannabis population due to hermaphrodite-induced cross-fertilization. Hermaphrodite cannabis also produces undesired seeds [13]. Therefore, the timely detection and elimination of both male and hermaphrodite plants are strongly desired in the cannabis industry.

Our group recently demonstrated that Raman spectroscopy (RS) can be used to differentiate between young male and female hemp plants [14]. This innovative optical sensing approach is based on the phenomenon of inelastic light scattering that occurs between incident photons and molecules present in the sample of interest [14,15]. As a result, photons with a change in the energies are scattered off the sample [16,17]. Acquisition of these photons allows for the direct chemical analysis of the sample. Using high-performance liquid chromatography (HPLC), our group showed that young male and female plants have statistically significant differences in the concentration of lutein [14]. This difference in the concentration of lutein in male vs. female hemp plants can be detected by RS enabling confirmatory differentiation between such plants. It should be noted that RS can be also used to detect and identify biotic and abiotic stresses in plants [18,19,20,21,22,23,24,25]. For instance, Sanchez and co-workers recently demonstrated that nitrogen, phosphorus, and potassium deficiencies, as well as salinity stresses, could be diagnosed in rice prior to their symptomatic appearances [24,26,27]. A growing body of evidence also suggests that RS can be used to phenotype species and their varieties and even differentiate between isogenic peanut varieties [28,29,30].

Expanding upon this, we investigate the extent to which RS can be used to identify hermaphrodite cannabis and differentiate between these species, male, and female plants. For this, we collected Raman spectra from hermaphrodite, male, and female plants of the same cannabis variety.

## 2. Materials and Methods

Plants: Cannabis plants (variety Futura 75) were grown in Longmont, CO in the greenhouse under 18 h of light during plant vegetation and 12 h during the stage of flowering. All plants were kept under the same vegetation conditions (light intensity, irrigation, and temperature).

Raman spectroscopy: Using a hand-held Resolve Agilent spectrometer equipped with an 831 nm laser, Raman spectra were collected from plant leaves with the following parameters: 1 s acquisition time, 495 mW power. Spectral baseline subtraction was performed by device software. On average, 2–3 spectra were collected from each plant at different locations and randomly selected leaves to ensure the robustness of the approach. We totally sampled 16–25 plants, collecting 50–77 spectra from each group (male, female, and hermaphrodites).

Chemometrics and machine learning: PLS_Toolbox was used to analyze all acquired spectra that were pre-processed by taking the 2nd derivative of all intensity values (2nd polynomial order and a filter length of 15) and then centered on the mean and median in Matlab. Using partial least squares discriminant analysis (PLS-DA), a true positive rate (TPR) was obtained for each category based on the accuracy rate of predictions of spectra to their category.

## 3. Results and Discussion

Spectroscopic signatures of hermaphrodite, male, and female hemp plants are comminated with vibrational bands that can be assigned to terpenes (796 cm^−1^), carotenoids (1002, 1115–1267, and 1525 cm^−1^), aromatic compounds (1609 cm^−1^), cellulose (746, 843, 917 and 1047 cm^−1^), proteins (1650–1680 cm^−1^), as well as aliphatic vibrations that cannot be assigned to any specific classes of biological molecules (1287–1439 cm^−1^), Figure 1 and Table 1 We found that the intensities of carotenoid vibrations were much more intense in the spectra collected from female plants compared to spectra collected from male plants. Finally, these bands exhibited the lowest intensities in the spectra collected from hermaphrodite cannabis. These findings demonstrate that hermaphrodite, male, and female hemp plants possess different amounts of carotenoids. Specifically, the concentration of carotenoids is the greatest in female cannabis, whereas hermaphrodite species demonstrate the lowest carotenoid content. These results are in good agreement with the research findings that were recently published by Higgins and co-workers [14]. The researchers found that female hemp plants possessed a substantially greater amount of lutein than male plants, which determined a drastic difference in the intensity of carotenoid bands in the spectra from female and male hemp plants [14].

Next, we investigated the extent to which difference in the intensity of carotenoid vibrations could be used to differentiate hermaphrodite, male, and female species. For this, we performed ANOVA of the intensity of three carotenoid vibrations at 1156, 1186, and 1218 cm^−1^, Figure 2. Our findings confirmed that the intensity of these carotenoid vibrations can be used as marker bands for quantitative differentiation between hermaphrodite, male, and female cannabis plants.

We also observed a new vibrational band in the spectra collected from hermaphrodites that was not evident in the spectra collected from both male and female cannabis. This band is centered around 1650 cm^−1^ and, therefore, can be assigned to amide I band of proteins [18,39]. It should be noted that amide I vibrations were also evident in the spectra collected from both male and female cannabis. However, in these spectra, these bands were found to be substantially red-shifted to ~1680 cm^−1^. The position of the amide I band can be used to interpret protein secondary structure. Unordered proteins typically exhibit amide I in around ~1680 cm^−1^, whereas the amide I band in the spectra collected from α-helical proteins is centered around 1650 cm^−1^ [18]. Based on these assignments, one can conclude that hermaphrodite cannabis possesses proteins with drastically different secondary structures compared to the proteins present in both male and female plants.

Finally, we used chemometrics to investigate the accuracy of differentiation between hermaphrodite, male, and female plants [40]. We built a binary PLS-DA model that demonstrates on average 99.6% accurate differentiation between these cannabis classes. Specifically, both male and female plants can be identified with 100% accuracy, whereas the identification of hermaphrodites can be achieved with 98.7% accuracy, Table 2 and Figure 3. These findings confirm that RS can be used for a robust and reliable differentiation between hermaphrodite, male, and female cannabis. It should be noted that although differentiation between male and female specimens can be achieved by subjective visual analysis of plants at the state of their flowering, this is a laborious and often not achievable task considering large agricultural areas used to produce cannabis. These agricultural areas require automated approaches that can be used for confirmatory differentiation between male and female plants prior to their flowering.

## 4. Conclusions

Our experimental results show that RS can be used for a label-free, non-invasive, and non-destructive differentiation between hermaphrodite, male, and female cannabis. This differentiation is based on differences in their biochemical profiles. Specifically, we found that female plants possess significantly higher amounts of carotenoids, whereas male plants have substantially lower concentrations of these important physiological compounds. Finally, hermaphrodite plants exhibit lower concentrations of carotenoids relative to both male and female plants. We also showed that coupling of RS with chemometrics and machine learning allows for the development of robust and reliable statistical algorithms that enable on average 99.6% accurate differentiation between male, female, and hermaphrodite cannabis. The portable nature of this analytical approach, as well as the intrinsic sensitivity of RS towards cannabinoid consent in cannabis, suggests that RS can be used directly in cannabis farms to control and monitor plant vegetation.

## Figures and Tables

**Figure 1 molecules-27-04978-f001:**
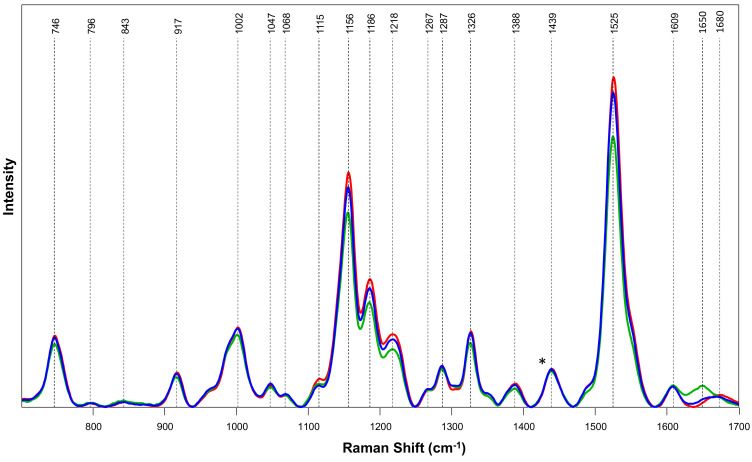
Averaged Raman spectra collected from leaves of male (**blue**), female (**red**), and hermaphrodite (**green**) plants. For each spectrum, 50–77 individual spectra collected from leaves of plants were averaged. Vibrational bands that correspond to certain chemicals present in the leaves are labeled and discussed in Table 1.

**Figure 2 molecules-27-04978-f002:**
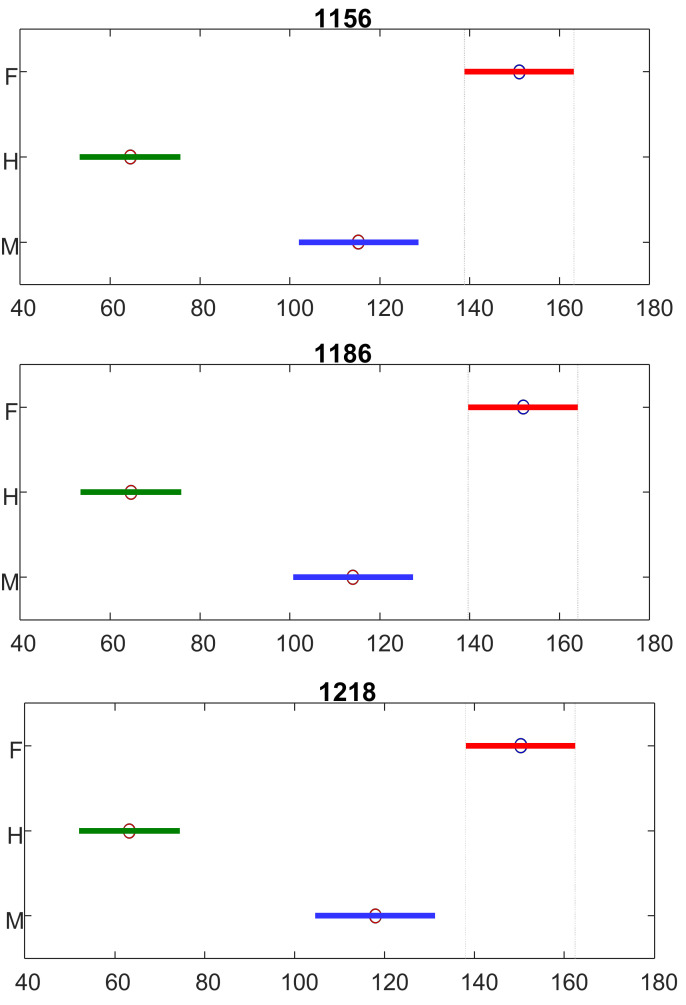
ANOVA of the intensity of 1156, 1186, and 1218 cm^−1^ bands demonstrate statistically significant differences in these vibrations that can be used as marker bands for differentiation between female (F), hermaphrodites (H), and male (M) cannabis plants. The ANOVA also reported a 95% confidence interval for the true value of median for each compared group. X axes represent ranks of 1156, 1186, and 1218 cm^−1^ band intensities (Dou, et. al., 2021).

**Figure 3 molecules-27-04978-f003:**
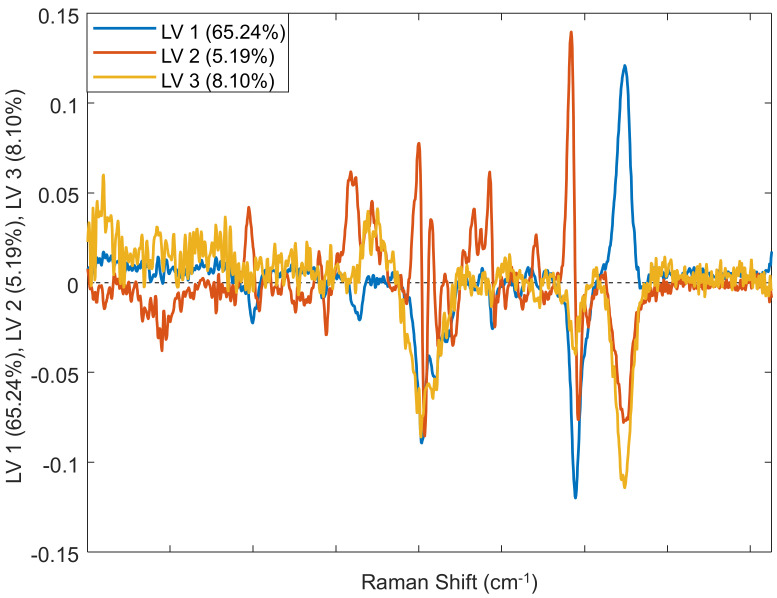
Loading plot of the three predictive components (PC) in the Raman spectra of male, female, and hermaphrodite cannabis plants. See Table 1 for a description of the biological origin of the bands.

**Table 1 molecules-27-04978-t001:** Assignments of vibrational bands observed in the spectra collected from the leaves of hemp plants.

Band	Vibrational Mode	Assignment
746	γ(C–O-H) of COOH	Pectin [31]
796	δ ring vibration	Terpenes [32]
843	ν(C-O-C)	Cellulose [33]
917	ν(C-O-C) In plane, symmetric	Cellulose, lignin [33]
1002	-C=C- (in plane)	Carotenoids [32]
1047–1068	ν(C-O) + ν(C-C) + δ(C-O-H)	Cellulose, lignin [33]
1115	-C=C- (in plane)	Carotenoids [33]
1156	-C=C- (in plane)	Carotenoids [32]
1186	ν(C-O-H) Next to aromatic ring + σ(CH)	Carotenoids [15]
1218	δ(C-C-H)	Carotenoids [15]
1267–1288	δ(C-C-H)	Aliphatics [34]
1326	δCH_2_ Bending	Aliphatics, cellulose, lignin [33]
1388	δCH_2_ Bending	Aliphatics [34]
1439	δ(CH_2_) + δ(CH_3_)	Aliphatics [34]
1525	-C=C- (in plane)	Carotenoids [35,36]
1609	ν(C-C) Aromatic ring + σ(CH)	Lignin [37,38]
1650–1680	Amide I	Proteins [18]

**Table 2 molecules-27-04978-t002:** Confusion table for spectra collected from hermaphrodite, male and female plants.

	Number of Spectra	TPR	Predicted as Female	Predicted as Male	Predicted as Hermaphrodite
Female	57	100%	57	0	0
Male	50	100%	0	50	0
Hermaphrodite	77	98.7%	0	1	76

## Data Availability

The data will be made available on reasonable request to the corresponding author.

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
