# Peer review of "Non-Invasive and Confirmatory Differentiation of Hermaphrodite from Both Male and Female Cannabis Plants Using a Hand-Held Raman Spectrometer"

_molecules, 2022, doi:10.3390/molecules27154978_

Round 1
Reviewer 1 Report
The manuscript reported a non-invasive method to distinguish the sexual identity of cannabis plants based on Raman spectroscopy. The idea may be interesting, but it is not convincing in its current form. I am not an expert of this plant, not sure whether the gender of the plant could be distinguished without the flowers. A google search seems suggested that the male cannabis, is commonly known as hemp while the female plant is usually refers to cannabis. It seems that the outlook appearance are quite different. The manuscript also suggested the appearance of the male and female plants is expressed in very early stages. Not sure whether that applies to hermaphroditic plant as well. If so, what is the point to develop an analytical method to distinguish the gender or sexual identity of the plants?
In developing classification models, sampling plan is critical to the reliability and effectiveness of the model. The sampling plan and spectra collection was not clearly described. It seems that only plants from one geographical site were used for study, this does not looks like a very representative samples. Did the spectrum being measured at different seasons? The manuscript just mentioned 2-3 spectra were collected for each plant. Were each spectrum collected on a same leaf of the plant? Any criteria on selection of the leaves? Would it be better to sample a few more leaves to get a representative average?
Fig 2 is presented in a way that is incomprehensible. It is the species verse intensity at different wavenumber. In Fig 1, it is quite clear that the peak at 1525 wavenumber is the strongest peak. Why it is not used in ANOVA? The average spectrum at 1156, 1186 and 1218 wavenumber do shown different intensity in Fig 1. Why the intensity (x-axis) shown in Fig 2 are the same at these three different wavenumber? BTW, I would expect the wavenumber to be put on the y-axis rather than x-axis.
Fig 1 already demonstrate quite clearly the spectra features that correlate with the sexual traits, not sure the purpose of Fig 3. I would expect a score plot of the PLS-DA analysis to see how scatter are the spectra, but it is not shown. Finally, the classification model should be validated.
Frankly speaking, I am disappointed with the quality of this manuscript.
Author Response
First, we want to thank the reviewer for provided suggestions. They helped to improve our work. Below are point-by-point answers to all raised concerns.
- The manuscript also suggested the appearance of the male and female plants is expressed in very early stages. Not sure whether that applies to hermaphroditic plant as well. If so, what is the point to develop an analytical method to distinguish the gender or sexual identity of the plants?
Response: visual differentiation between male and female plants requires advanced botanical expertise that cannot be possessed by most of farmers and plant breeders that grown cannabis. Therefore, there is an on-going search for the automated approach that should ideally be non-invasive and chemical free. Raman spectroscopy (RS) meets these criteria. This work showed that RS can distinguish between male, female and hermaphrodite cannabis based on their spectroscopic signatures. Thus, RS can be used to enable automated plant sexing which is vitally important since one male or hermaphrodite plant in the farm or a greenhouse can cause detrimental effect to the harvested female plants.
- In developing classification models, sampling plan is critical to the reliability and effectiveness of the model. The sampling plan and spectra collection was not clearly described. It seems that only plants from one geographical site were used for study, this does not looks like a very representative samples. Did the spectrum being measured at different seasons? The manuscript just mentioned 2-3 spectra were collected for each plant. Were each spectrum collected on a same leaf of the plant? Any criteria on selection of the leaves? Would it be better to sample a few more leaves to get a representative average?
Response: we apologize for the lack of clarity in the provided description of the sampling approach. We modified the section accordingly:
“On average, 2-3 spectra were collected from each plant at different locations and randomly selected leaves to ensure robustness of the approach. We totally sampled 16-25 plants collecting 50-77 spectra from each group (male, female and hermaphrodites).”
Thus, 16-25 plants were scanned at randomly chosen location on their leaves. We collected 1-2 spectra per leaf. We purposely did not have any criteria for the leaf selection to ensure robustness of the approach.
The reviewer is correct about one geographical location of the experiment. It should be noted that all farmers in the US strongly resist growing any male plants. Therefore, it took us more than 2 years to identify the farmer who provided the access to his facility. Finally, since plants are grown in the greenhouse under 18 hours of light during plant vegetation and 12 hours, vegetation season is not important.
- Fig 2 is presented in a way that is incomprehensible. It is the species verse intensity at different wavenumber. In Fig 1, it is quite clear that the peak at 1525 wavenumber is the strongest peak. Why it is not used in ANOVA? The average spectrum at 1156, 1186 and 1218 wavenumber do shown different intensity in Fig 1. Why the intensity (x-axis) shown in Fig 2 are the same at these three different wavenumber? BTW, I would expect the wavenumber to be put on the y-axis rather than x-axis.
Response: we want to point out that all wavelengths (1156, 1186 and 1218, as well as 1525) originate from the same class of molecules, carotenoids. Our results simply showed that any of those wavelengths could be picked up for differentiation between M, F and H plants.
We modified figure caption to explain meaning of X axes:
“The ANOVA also reported a 95% confidence interval for the true value of median for each compared group. X axes represent ranks of 1156, 1186 and 1218 cm-1 band intensities (Dou, et. al. 2021). “
Fig 1 already demonstrate quite clearly the spectra features that correlate with the sexual traits, not sure the purpose of Fig 3. I would expect a score plot of the PLS-DA analysis to see how scatter are the spectra, but it is not shown. Finally, the classification model should be validated.
Response: we provided loading plot of PLS-DA model in Figure 3. This figure demonstrates biological significance of the LVs used in PLS-DA. Thus, it shows bands that were important for M/F/H identification.
We modified the figure caption accordingly: “See Table 1 for description of biological origin of the bands. “
Reviewer 2 Report
he present manuscript is close from the previous published manuscript
Higgins S, Jessup R, Kurouski D. 2022. Raman spectroscopy enables highly accurate differentiation between young male and 232 female hemp plants. Planta. Mar 13;255:85
Several phrases were close in both publications.
In both paper table 1 was the same.
The previously published manuscripts differentiate male from female plants.
The present manuscript differentiates, male, female, and hermaphrodite plants.
Poor information was provided about the machine learning procedure described in the manuscript. The whole data set must be provided. Authors must provide a Microsoft Excel file containing all original Expectra.
The hermaphrodite plants may be differentiated from male and female plats just using the absorption band in 1650. Why to use chemometrics?
The method description is poor, how many plants from each species were analyzed? What was the data set used to build the machine learning model?
For these reasons, I recommend rejection.
Author Response
Poor information was provided about the machine learning procedure described in the manuscript. The whole data set must be provided. Authors must provide a Microsoft Excel file containing all original Expectra.
Response: we cannot provide experimental spectra due to the signed agreement with the sponsor.
The hermaphrodite plants may be differentiated from male and female plats just using the absorption band in 1650. Why to use chemometrics?
Response: we infer that analysis of intensity of one band may not be sufficient, therefore, to ensure robustness of the approach, we used chemometrics.
The method description is poor, how many plants from each species were analyzed? What was the data set used to build the machine learning model?
Response: we modified the materials and methods section accordingly:
“On average, 2-3 spectra were collected from each plant at different locations and randomly selected leaves to ensure robustness of the approach. We totally sampled 16-25 plants collecting 50-77 spectra from each group (male, female and hermaphrodites). “
Reviewer 3 Report
Dear authors,
You decribed the ''Non-Invasive and Confirmatory Differentiation of Hermaphrodite from Both Male and Female Cannabis Plants Using a Hand-Held Raman Spectrometer''
I found this work extremely interesting and my suggestion is that you paper should be accepted after minor revision. In the pdf below you will find some correction and some suggestions to improve your work.
Congratulations for your work and I am sure that experiments like yours must be done to give valuable information about the plants.
Thank you

Author Response
We are grateful to the reviewer for ranking our work so high and for the positive feedback about the experimental results.
Round 2
Reviewer 1 Report
The revised manuscript did not address the previous comments satisfactory. In the response, the authors said identifying the sexual identity of the plants is a difficult and important problems, but in the sample collection, they suggested not many US farmer would grow like to allow male or hermaphroditic plants. Does that suggest that the farmer do have a way to identify the non-female plants and eliminate them during the process?
In developing classification models, sampling plan is critical to the reliability and effectiveness of the model. The authors require to justify the choice of only one single farm. Even a single farm is chosen, what is the background of the farm, the geographical size? The production size? The sample collections time? How many growing seasons that the samples involved? 16-25 samples appeared to be small.
Fig 2 is presented in a way that is incomprehensible. It is the species verse intensity at different wavenumber. In Fig 1, it is quite clear that the peak at 1525 wavenumber is the strongest peak. Why it is not shown in Fig ? Why spectral intensities is not used in Fig 2? I would expect the spectral intensities served as the depending variable showing on the y axis while sexual identity as the independent variable showing on the x axis.
Fig 1 already demonstrate quite clearly the spectra features that correlate with the sexual traits, not sure the purpose of Fig 3. I would expect a score plot of the PLS-DA analysis to see how scatter are the spectra, but it is not shown. Finally, the classification model should be validated. I didn’t see any additional information given that can be given by Fig 3 when there is a Fig 1. Fig 1 plus ANOVA already clearly shown the spectra region showing differences of the sexual identity of the plant. Finally, x-axis of Fig 3 has no number label to show the wavenumber of the features.
Reviewer 2 Report
The manuscript is professionally written and organized. However, the authors must provide all Raman spectra used in the manuscript was a Microsoft Excel file.